# Influence of Thermal Treatment on SCC and HE Susceptibility of Supermartensitic Stainless Steel 16Cr5NiMo

**DOI:** 10.3390/ma13071643

**Published:** 2020-04-02

**Authors:** Linda Bacchi, Fabio Biagini, Serena Corsinovi, Marco Romanelli, Michele Villa, Renzo Valentini

**Affiliations:** 1Letomec srl, 56126 Pisa, Italy; l.bacchi@letomec.com (L.B.); f.biagini@letomec.com (F.B.); research@letomec.com (S.C.); 2Baker Hughes, 50127 Firenze, Italy; marco.romanelli@bhge.com; 3Department of Civil and Industrial Engineering, Pisa University, 56122 Pisa, Italy; renzo.valentini@unipi.it

**Keywords:** supermartensitic stainless steels, sour environment, stress corrosion cracking, hydrogen embrittlement, tempering temperature, retained austenite, instantaneous strain hardening coefficient

## Abstract

A 16Cr5NiMo supermartensitic stainless steel was subjected to different tempering treatments and analyzed by means of permeation tests and slow strain rate tests to investigate the effect of different amounts of retained austenite on its hydrogen embrittlement susceptibility. The 16Cr5NiMo steel class is characterized by a very low carbon content. It is the new variant of 13Cr4Ni. These steels are used in many applications, for example, compressors for sour environments, offshore piping, naval propellers, aircraft components and subsea applications. The typical microstructure is a soft-tempered martensite very close to a body-centered cubic, with a retained austenite fraction and limited *δ* ferrite phase. Supermartensitic stainless steels have high mechanical properties, together with good weldability and corrosion resistance. The amount of retained austenite is useful to increase low temperature toughness and stress corrosion cracking resistance. Experimental techniques allowed us to evaluate diffusion coefficients and the mechanical behaviour of metals in stress corrosion cracking (SCC) conditions.

## 1. Introduction

Supermartensitic stainless steels have been developed to offer increased corrosion resistance, especially in sour environments (containing hydrogen sulphide H_2_S) [1,2,3]. They also present remarkable weldability and fracture toughness even at low temperatures, in comparison with traditional martensitic stainless steels. Thanks to all these characteristics, they are suitable for various applications such as compressor impellers, offshore piping, naval propellers and aircraft components.

Corrosion resistance and ductility can be improved by reaching a homogeneous tempered martensitic structure with a certain amount of retained austenite. The martensitic transformation end temperature is typically close to room temperature [4,5,6], which is why a sub-zero cooling is often performed to avoid excessive retained austenite formation.

The high nickel content reduces the austenitizing temperature Ac_1_, therefore a low tempering temperature must be chosen to avoid the excessive formation of reversion austenite *γ_rev_*. Reversion austenite can become unstable during cooling and transform into fresh martensite (instead of retained austenite), implying a hardness increase, which is usually not acceptable [7]. On the other hand, the retained austenite fraction is stabilized by carbon and manganese enrichment.

Retained austenite, in martensitic or ferritic matrices, can also turn into fresh martensite in the case of plastic straining, and this microstructural modification can play an important role in steel hydrogen embrittlement resistance. Karlsen et al. recently observed this phenomenon in supermartensitic stainless steels [1,8]. Hydrogen is much more soluble in an austenitic lattice and, as consequence of the microstructure transformation, the same amount of hydrogen gathers into a martensitic microstructure. The lower matrix solubility leads to critical cohesion stress reduction, enhancing hydrogen-induced cracking nucleation.

There are three uppermost mechanisms in hydrogen embrittlement failures. The hydrogen-enhanced decohesion (HEDE) is associated with local hydrogen accumulation which reduces the cohesive strength. It generally gives rise to a brittle fracture without any local deformation. Hydrogen-enhanced local plasticity (HELP) instead consists of hydrogen promotion of planar slips and dislocation motion, showing localized plastic deformation and slip-band cracking. Finally, the adsorption-induced dislocation emission (AIDE), is where adsorbed hydrogen is responsible for dislocation emission towards a crack tip and the cracking is nanocleavage-like [2,9,10].

A sour environment establishes a heavy working condition where hydrogen embrittlement and stress corrosion cracking can occur, acting as a possible cause for component failure [11]. This specific service case operation is covered by NACE (National Association of Corrosion Engineers) reference standards [12], which impose severe restrictions on material properties. A maximum hardness value equal to 23 HRC was imposed for class 13Cr4Ni steels. For 16Cr5NiMo, the industrial hardness limit is currently 28 HRC. Some restrictions on yield strength and ultimate tensile strength are also defined as 620 MPa and 880 MPa, respectively.

It is not simple to meet these requirements, since a further carbon content reduction is not suitable because of the consequent reduction in all mechanical properties to values lower than required, thus an accurate definition of thermal treatment is needed [13].

Thermal treatment consisted of austenitization, oil quenching and a sub-zero cooling. Afterwards, the material was subjected to a double tempering treatment. A typical microstructure is mainly tempered martensite (*α’*), with a lattice constant close to 1 because of the very low carbon content, and very similar to ody centered cubic (BCC). A certain amount of retained austenite (*γ_r_*) is present, a function of the chemical composition and treatment conditions (heating rate, dwell time at a high temperature, chromium carbides precipitation, alloy elements redistribution and cooling rate). Furthermore, a small quantity of *δ* ferrite can form.

Many authors investigated the comparison between the reversion austenite formed during tempering treatment and the effective retained austenite fraction, which remains stable at room temperature [7,14,15]. It was found that the reversed austenite fraction formation continuously increased with tempering temperature, but retained austenite was not a monotonic function of tempering temperature and presents a maximum [16]. This can be explained by considering nickel depletion in the austenitic phase as a consequence of reversed austenite formation, which leads to an increase in the martensite transformation start temperature, enhancing partial transformation into fresh martensite during cooling. Since the presence of fresh martensite in the final microstructure is an undesirable condition because of the consequent excessive hardness (out of standard), a second tempering treatment is usually performed.

The present study was aimed at investigating the influence of different tempering temperatures on 16Cr5NiMo steel susceptibility to hydrogen embrittlement. A series of permeation tests and slow strain rate tests were performed, to evaluate the influence of different amounts of retained austenite in the microstructure. Moreover, a study on instantaneous strain hardening coefficients was carried out.

## 2. Materials and Methods

Supermartensitic stainless steel class 16Cr5NiMo is characterized by the chemical composition reported in Table 1.

Samples were obtained by hot-forged rotors for centrifugal compressors. Considering that chromium carbides Cr_23_C_6_ formed during stress relieving after forging are responsible for stainless steel’s sensitization to corrosion, a solubilization treatment was performed to allow their dissolution. A typical tempering process is carried out at 630 °C, but the temperature can be varied due to mechanical requirements. In the present work, three different tempering temperatures were applied after quenching in order to obtain different amounts of retained austenite in the final microstructure [14]. 

The investigated material was the same as that of De Sanctis et al., thus austenite fractions were deduced according to [16] with reference to the alloy A results. The steels’ identifying letters and thermal treatment conditions are summarized in Table 2.

The differences in terms of austenite fractions were expected to modify the steels’ susceptibility to stress corrosion cracking and hydrogen embrittlement. Thanks to the face-centered cubic (FCC) lattice, with elevated hydrogen solubility, the austenite phase works as a hydrogen trap, avoiding further diffusion towards cracks or other critical points [17,18].

### 2.1. Metallography

Samples from all treatment groups were prepared for optical (Leica DMI 300M, Wetzlar, Germany) and scanning electron microscopies, SEM (JEOL JSM 5600LV, Akishima, Japan).

After polishing, Beraha CdS etching was used for light microscope images. The presence of carbides, highlighted in purple-blue colours by etching, was evident for all treatment conditions. They were principally precipitated in correspondence with grain boundaries and are indicated by the red arrows in the figures (Figure 1a, Figure 2a and Figure 3a), while SEM images (Figure 1b, Figure 2b and Figure 3b) reported precipitated details with a higher magnification. Metallographic analysis showed a grain size within the range of 50–150 μm.

### 2.2. Permeation Tests

Test samples consisted of metal sheets 50 × 50 mm, 0.5 mm thick. They were subjected to hydrogen permeation after a prior mechanical surface cleaning by means of abrasive papers and washed with an ultrasonic cleaner with acetone. A test solution with H_2_SO_4_ 1 N + As_2_O_3_ 10 mg/L in distilled water was used and a current density of 1 mA/cm^2^ was applied for all three materials. An Ag/AgCl reference electrode was used for voltage measurements. A nitrogen purging flux was applied inside the electrochemical cell to avoid any sample surface oxidation during the test execution.

The diffusion coefficient was calculated by the time lag method according to the ISO 17081:2014 standard [19]; see Equation (1). Once calculated, the permeation flux integral curve, *t_L_*, was determined as the abscissae intercept of the steady state tangent line, while *l* was the sample thickness.
(1)$$D=\frac{l^{2}}{6t_{L}}$$

### 2.3. Slow Strain Rate Tests

Mechanical testing was carried out on as-received samples to determine the reference behaviour of each treatment condition after hydrogen charging by means of MTS CERT machine 647 All-temperature Hydraulic Wedge Grips (equipped with dedicated autoclave). The samples’ geometry was in agreement with NACE TM0177 and dedicated threaded gripping ends were machined (Figure 4). Tests were performed with reference to NACE TM0198 and ASTM G129 [20,21], and charging methods are reported in Section 2.3.1.

The test results were finally analyzed in terms of embrittlement index, calculated with reference to the samples’ elongation at break, Equation (2). A reduction of 30% was considered as the threshold value in agreement with reference standard ASTM STP 962 [22].
(2)$$F_{A}\%=\frac{A_{AR}-A_{H}}{A_{AR}}\times 100$$

#### 2.3.1. Hydrogen Charging

Some samples were subjected to mechanical testing after electrochemical hydrogen charging in a cathodic protection condition (scheme reported in Figure 5), simulating subsea applications. Test specimens were electrically connected to a magnesium sacrificial anode in a saline solution (3.5% NaCl). The electrochemical potential difference between Mg and stainless steel gives rise to a current, because of the galvanic corrosion setup. The voltage between the anode and the cathode of the galvanic coupling was measured by means of a reference electrode and corresponded to 1.96 V. The test temperature varied from room temperature to 60 °C by means of a thermostatic bath.

Other samples were subjected to slow strain rate in an autoclave in order to simulate hydrogen absorption during operation, as in the external hydrogen embrittlement phenomenon. Specimens were immersed in NACE A solution during mechanical testing. Standard conditions were applied (p_H_ = 3.5, hydrochloric acid [Cl^−^] = 100 ppm, buffer CH_3_COONa, p_tot_ = 1 bar, p_H2S_ = 10% p_tot_, room temperature) [23].

### 2.4. Instantaneous Strain Hardening Coefficient 

By means of a rheological model with an instantaneous strain hardening coefficient, the plastic strain range was analytically described. The main aim of this investigation was to evaluate the possibility of observing the transformation of plastic strain-induced austenite to fresh martensite. Considering the study of Dimatteo [24], some materials present an inflection point in the strain hardening trend, thus an experimental data regression with a fourth grade polynomial was made to eventually appreciate this feature (Equation (3)).
(3)$$ln\left(\sigma_{t}\right)=\sum_{i=0}^{4}A_{i}ln^{i}\left(\varepsilon_{pt}\right)$$

The best fitting parameters and instantaneous strain hardening coefficient (Equation (4)) were evaluated for all materials.
(4)$$n=\frac{d\left(ln\left(\sigma_{t}\right)\right)}{d\left(ln\left(\varepsilon_{pt}\right)\right)}$$

## 3. Results

### 3.1. Permeation Tests

Effective diffusion coefficients were calculated according to the time lag method (see Section 2.2) and the results are reported in Table 3. For completeness, permeation flux curves are plotted in Figure 6.

### 3.2. Slow Strain Rate Tests

The slow strain rate test results are reported in Table 4, two specimens were tested in as-received conditions with different strain rates.

In Figure 7, embrittlement indexes determined on the samples’ elongation at break, as described in Equation (2), were correlated to the hydrogen charging method to compare the behaviour of groups I, X and O.

#### SEM Fractographic Analysis

Considering the test conditions described in dedicated standard references, with the sample immersed in NACE A solution, fracture surfaces were observed by a scanning electron microscope to evaluate the main fracture mode. Images related to all three material groups are reported in Figure 8, Figure 9 and Figure 10.

Samples of group I showed intergranular brittle fractures with some secondary cracks and some transgranular areas. For groups X and O, larger transgranular quasi-cleavage areas were found. The fracture mode therefore appears to be correlated with the amount of retained austenite and quasi-cleavage was dominant for higher austenite amounts [2,25].

### 3.3. Instantaneous Strain Hardening Coefficient

Polynomial regression best-fitting parameters are reported in Table 5.

## 4. Discussion

The experimental work was done in order to differentiate and classify the microstructure obtained with various tempering temperatures in terms of stress corrosion cracking and hydrogen embrittlement susceptibility.

The final microstructure of supermartensitic stainless steel 16Cr5NiMo was biphasic (martensite–austenite). Previous studies by means of TEM (Transmission Electron Microscope) showed the austenite fraction as elongated interlath films in a martensite matrix (lattice constant close to 1 as a body-centered cubic because of the low carbon content) [16].

Permeation tests were carried out at room temperature using a patented new technology and method [26]. The equipment was based on a solid-state gas sensor and was able to perform fast permeation of sheet metal samples, evaluating the effective hydrogen diffusion coefficient [27,28]. 

The permeation test time, and consequently the diffusivity of metal, is a function of temperature and microstructure. For example, it is lower in austenitic stainless steels (FCC) compared to ferritic stainless steels (BCC): it is D_FCC_ ≈ 10^−16^ m^2^/s and D_BCC_ ≈ 10^−12^ m^2^/s, respectively [29,30]. Thus, an austenite lattice in a martensitic matrix can be considered a hydrogen trap.

The microstructures of the steels were different in terms of retained austenite amounts, and a variation in hydrogen diffusivity was found [31]. Consequently, confirming the other literature studies, the hydrogen diffusion coefficient has a certain proportionality with the austenitic phase content. The martensite tempering temperature and eventual carbides precipitation may have also had an effect. A good exponential correlation was found including the data of the supermartensitic stainless steel under investigation and generic ferritic and austenitic stainless steel values (Figure 11).

The hydrogen diffusivity and plastic behavior of the steels under investigation were clearly influenced by the presence of a certain amount of retained austenite. The higher the fraction of the austenitic phase, the lower the effective hydrogen diffusivity of the steel. Group O, with a second tempering temperature of 670 °C and containing the maximum quantity of retained austenite (according to [16]), showed the lowest diffusivity [30].

Considering that in the real operative life of typical components realized in 16Cr5NiMo, the yield strength of metal is usually never exceeded and no plastic strain-induced transformation into martensitic microstructure is expected, retained austenite acts as a trap for hydrogen. In fact, the hydrogen accumulation in the austenitic phase prevents it from moving towards the crack tip, with a consequent increase in hydrogen embrittlement resistance [32,33]. A different evolution is possible for the parts which work in a plastic strain field.

It was also evident that steel’s susceptibility to hydrogen embrittlement was more influenced by the specific hydrogen charging method than the amount of retained austenite, especially because a big number of precipitates were found at the grain boundaries. Chromium carbides, M_23_C_6_ type are typical, as reported in [16], where Alloy A corresponds to the same material as in the present work. 

Finally, considering the instantaneous strain hardening coefficient, a correlation between austenite presence and plastic behavior is evident, but its role should be investigated more deeply. 

The true stress–true strain curve was calculated from tensile test machine data. A plastic deformation range, up to necking initiation, was selected and reported in a bi-logarithmic diagram. A polynomial regression curve of group X is reported in Figure 12a as an example. Figure 12b presents instead the comparison between instantaneous strain hardening coefficients calculated by a derivative of the polynomial (orange) and a finite difference method on experimental data (blue) of the same steel. The expected trend is very close to the calculated curve.

In Figure 13, the comparison between the as-received material and hydrogen charged samples was reported for each material group.

The first results showed that a major retained austenite content (as for materials X and O), makes a negligible hydrogen absorption influence on instantaneous strain hardening. The instantaneous strain hardening curve is probably mainly correlated to and dependent on austenite transformation into fresh martensite induced by plastic straining.

To summarize, the principal causes were both large grain size and precipitates at grain boundaries, and thus were mainly correlated to high temperature processes rather than tempering treatment variations. In fact, in all specimens, hydrogen mostly diffused through grain boundaries, showing intergranular brittle surfaces with numerous secondary cracks. All three examined variants of 16Cr5NiMo steel, independently from tempering treatment, resulted in susceptibility to stress corrosion cracking both in the case of SSR testing after hydrogen pre-charging (internal hydrogen embrittlement) and SSR testing of immersion in NACE solution (external hydrogen charging). Few differences were observed between as-received and charged materials; the major effect was the embrittlement behavior of the BCC matrix. The retained austenite effect was limited; this is correlated to the low values of the retained austenite percentages in the microstructure; between the three steels, there were small differences.

## 5. Conclusions

The results obtained in the present work prove that different tempering processes cause the formation of different amounts of retained austenite in 16Cr5NiMo steels, which give rise to complex phenomena:-Retained austenite has a role in the effective hydrogen diffusion coefficient, together with other significant parameters such as the martensite tempering process and secondary phases precipitations (carbides);-Retained austenite transformation due to plastic strain flow, which can be appreciated by means of mechanical tests;-The embrittlement effect, at least for the investigated conditions, appears to be not strictly correlated with the retained austenite amounts, but mainly to large dimensions of prior austenitic grain size.

The investigation techniques used in this study allowed for very long permeation times, and thus they are useful for studying the behaviour of supermartensitic stainless steels. However, more microstructural and hydrogen embrittlement susceptibility investigations are needed, and it would be desirable to test materials with a finer grain size, where the damage effect could be correlated to the presence of retained austenite amounts.

## Figures and Tables

**Figure 1 materials-13-01643-f001:**
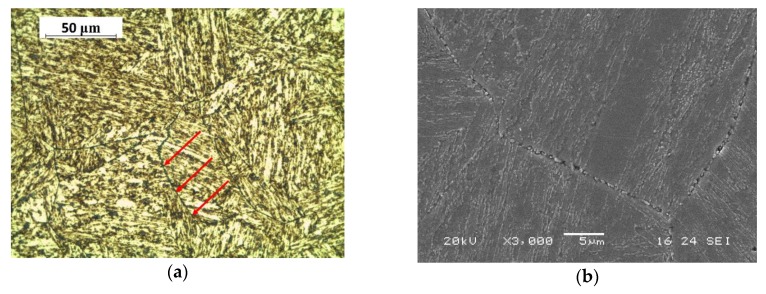
(**a**) Light microscope image (red arrows indicate carbides) and (**b**) SEM image of group I steel microstructure.

**Figure 2 materials-13-01643-f002:**
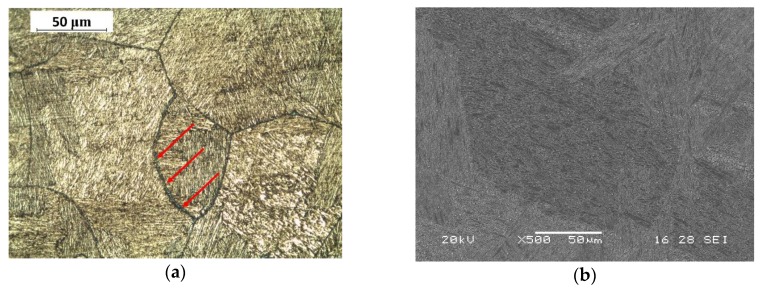
(**a**) Light microscope image (red arrows indicate carbides) and (**b**) SEM image of group X steel microstructure.

**Figure 3 materials-13-01643-f003:**
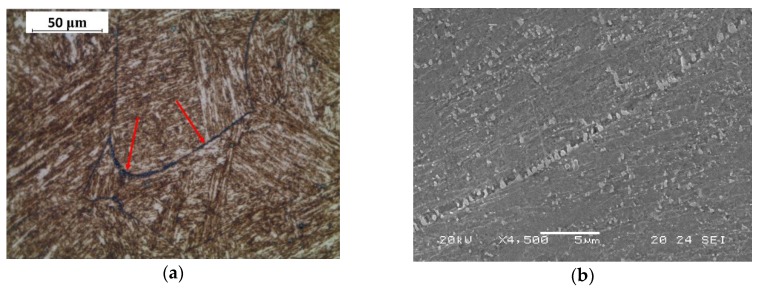
(**a**) Light microscope image (red arrows indicate carbides) and (**b**) SEM image of group O steel microstructure.

**Figure 4 materials-13-01643-f004:**
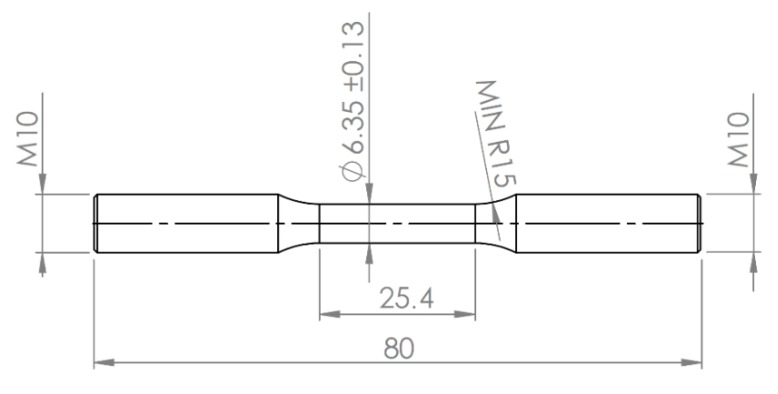
Specimen geometry with reference to NACE TM0177.

**Figure 5 materials-13-01643-f005:**
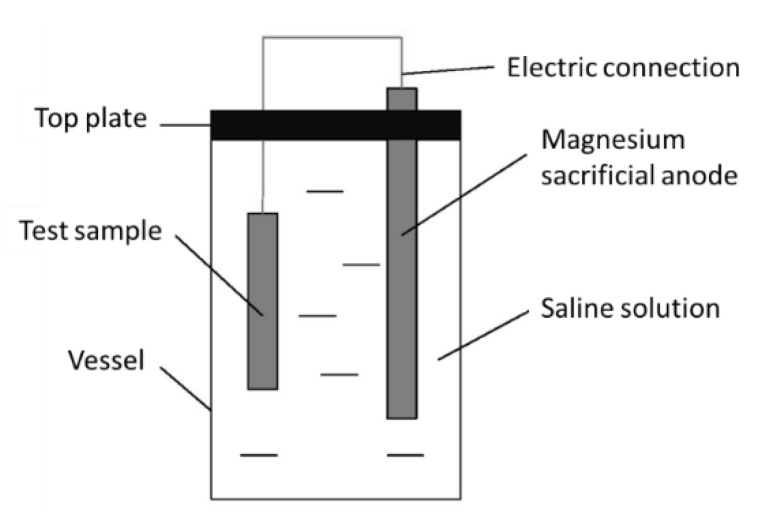
Electrochemical charging scheme.

**Figure 6 materials-13-01643-f006:**
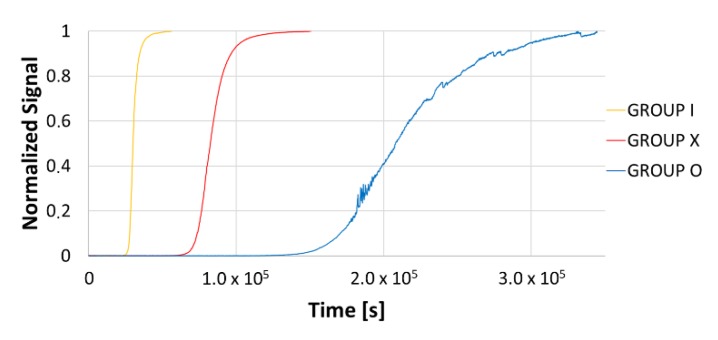
Permeation flux curves for the steels under investigation.

**Figure 7 materials-13-01643-f007:**
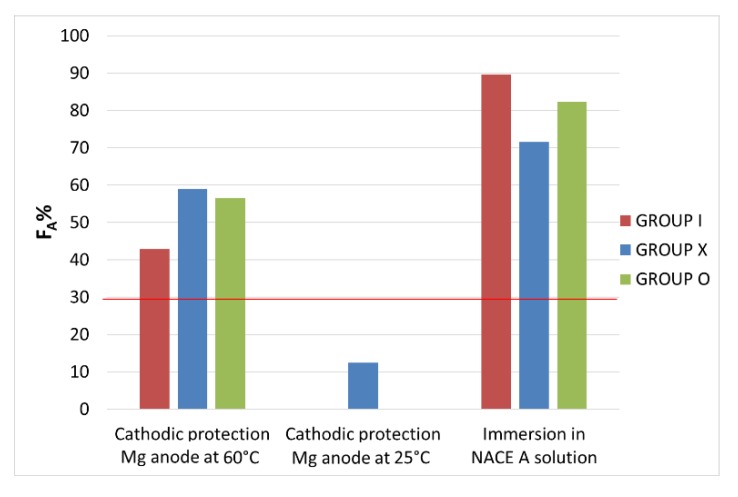
Embrittlement index calculated on elongation at break. The charging method is reported on the bottom and the red line corresponds to the threshold value.

**Figure 8 materials-13-01643-f008:**
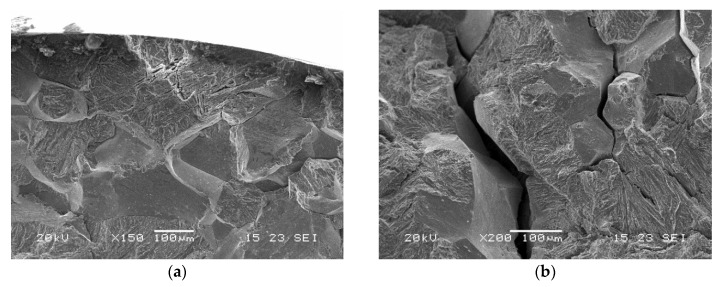
SEM images of fracture surfaces of a group I sample after SSRT test of immersion in NACE A solution (**a**) edge, (**b**) core.

**Figure 9 materials-13-01643-f009:**
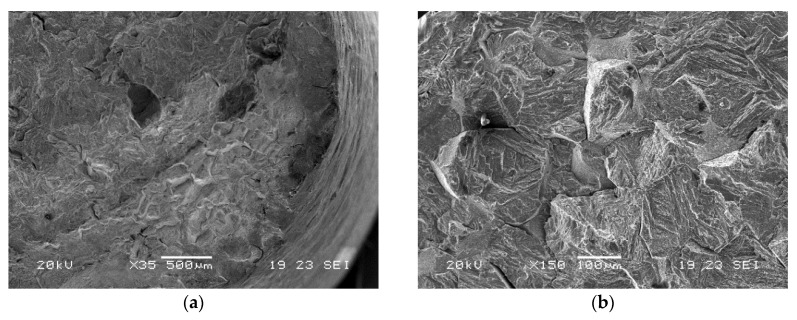
SEM images of fracture surfaces of a group X sample after SSRT test of immersion in NACE A solution (**a**) edge, (**b**) core.

**Figure 10 materials-13-01643-f010:**
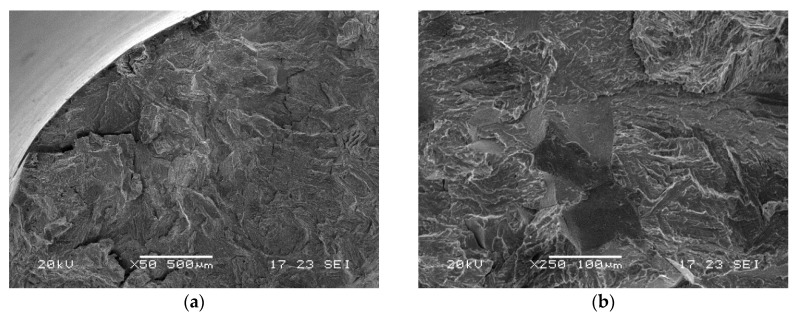
SEM images of fracture surfaces of a group O sample after SSRT test of immersion in NACE A solution (**a**) edge, (**b**) core.

**Figure 11 materials-13-01643-f011:**
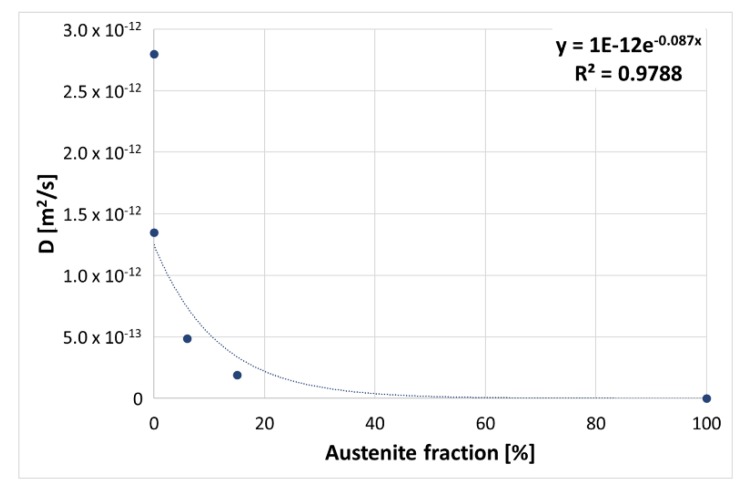
Exponential regression of the hydrogen diffusion coefficient as a function of the austenitic phase fraction.

**Figure 12 materials-13-01643-f012:**
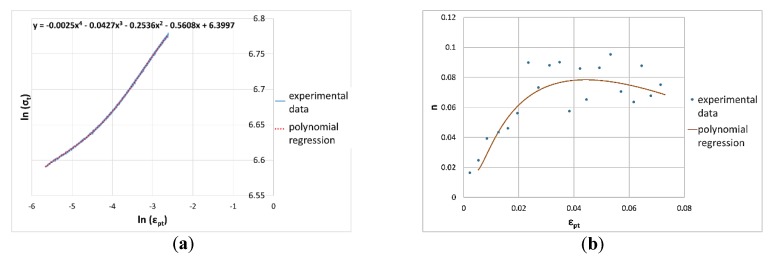
Example of (**a**) true stress-true strain plastic range up to necking initiation in a bi-logarithmic diagram and regression with fourth degree polynomial, (**b**) instantaneous strain hardening coefficient comparison between experimental data and polynomial regression on group X.

**Figure 13 materials-13-01643-f013:**
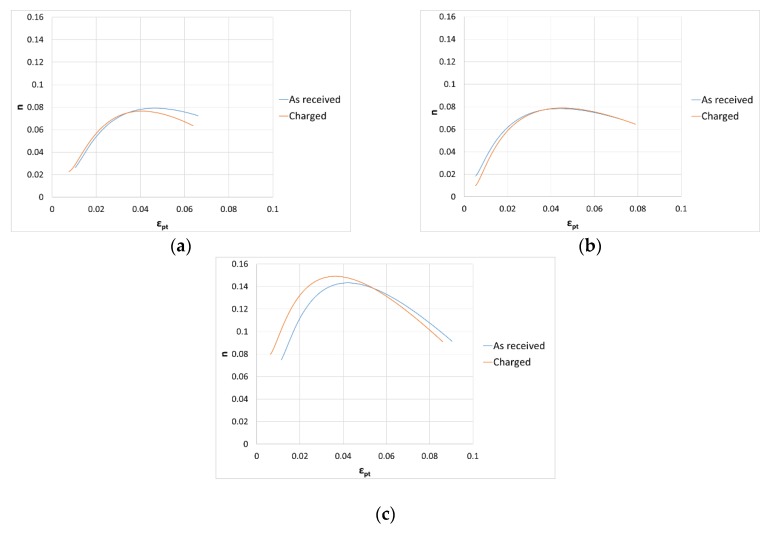
Instantaneous strain hardening coefficient as a function of plastic strain. Comparison between as-received material and after hydrogen charging under Mg cathodic protection at 60 °C for (**a**) group I, (**b**) group X and (**c**) group O.

**Table 1 materials-13-01643-t001:** Chemical composition of analyzed 16Cr5NiMo steels.

**% C**	**% Cr**	**% Ni**	**% Mo_max_**	**% S_max_**	**% P_max_**	**% V_max_**
0.035	16.10	4.55	0.12	0.003	0.012	0.035
**% Ti_max_**	**% N**	**% Mn_max_**	**% Si_max_**	**% Al_max_**	**% Cu_max_**	**-**
0.012	0.028	0.65	0.050	0.035	0.11	-

**Table 2 materials-13-01643-t002:** Thermal treatment conditions for the steels under investigation.

ID	Solubilization	Quenching Medium and Temperature	I Tempering	II Tempering	Cooling Medium	Retained Austenite (vol%)
I	1020 °C - 2 h	Sub-zero oil quench	580 °C - 2 h	540 °C - 2 h	air	0
X	1020 °C - 2 h	Sub-zero oil quench	630 °C − 2 h	540 °C - 2 h	air	6
O	1020 °C - 2 h	Sub-zero oil quench	670 °C − 2 h	540 °C - 2 h	air	15

**Table 3 materials-13-01643-t003:** Permeation test results for the steels under investigation.

ID	t_L_ (s)	D (m^2^/s)
I	39,219	1.35 · 10^−12^
X	85,381	4.88 · 10^−13^
O	220,440	1.89 · 10^−13^

**Table 4 materials-13-01643-t004:** SSR test conditions and results.

ID	Test Condition	Rm (MPa)	A (%)	RA (%)	F_A_ (%)	F_RA_ (%)
I	As received ($\dot{\varepsilon }$ = 10^−4^ s^−1^)	835	24.1	73.1	-	-
	As received ($\dot{\varepsilon }$ = 10^−6^ s^−1^)	826	19.7	^1^	-	-
	Cathodic protection Mg anode 60 °C - 5 days	825	13.8	29.8	43	59
	Immersion NACE A	795	2.5	17.2	90	76
X	As received ($\dot{\varepsilon }$ = 10^−4^ s^−1^)	817	24.6	67.1	-	-
	As received ($\dot{\varepsilon }$ = 10^−6^ s^−1^)	820	22.9	^1^	-	-
	Cathodic protection Mg anode 25 °C - 3 days	799	21.5	48.8	12	27
	Cathodic protection Mg anode 60 °C - 5 days	837	10.1	22.8	59	66
	Immersion NACE A	809	7	13.1	72	80
O	As received ($\dot{\varepsilon }$ = 10^−4^ s^−1^)	801	28.3	74.8	-	-
	As received ($\dot{\varepsilon }$ = 10^−6^ s^−1^)	804	19.3	^1^	-	-
	Cathodic protection Mg anode 60 °C - 5 days	810	12.3	24.8	57	67
	Immersion NACE A	752	5	0.17	82	100

^1^ Measures not available.

**Table 5 materials-13-01643-t005:** Instantaneous strain hardening regression best-fitting parameters, for materials in as-received condition.

ID	A_0_	A_1_	A_2_	A_3_	A_4_
I	−0.0038	−0.0621	−0.3591	−0.8098	+6.2037
X	−0.0025	−0.0427	−0.2536	−0.5608	+6.3997
O	−0.0054	−0.0908	−0.5438	−1.2557	+5.8164

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
