# Peer review of "Influence of Thermal Treatment on SCC and HE Susceptibility of Supermartensitic Stainless Steel 16Cr5NiMo"

_materials, 2020, doi:10.3390/ma13071643_

Round 1

Reviewer 1 Report

The manuscript entitled „Influence of thermal treatment on SCC and HE susceptibility of supermartensitic stainless steel 16Cr5NiMo“ deals with hydrogen embrittlement. This phenomenon is very critical in structural components from the reliability point of view. However, the present form of the manuscript is not suitable for publication without major changes.

Main comments and recommendations:

  1. The paper should keep the structure where the Materials and Methods chapter should contain a full description of all experimental methods and parameters used together with initial material characteristics (microstructural and mechanical). For example “Slow Strain Rate tests” is not described. The literature research should be elaborated rather in introduction or discussion. The Discussion here totally misses the literature context and Conclusions are not present at all. This manuscript organization should be deeply reorganised.
  2. All symbols used in the equations should be explained – maybe a list of symbols will be beneficial.
  3. It is not clear why authors use polynomial fit with 4th order when in work [27] is used just 3rd order? Additionally, ones are used ln (eq 5 in [27]) and once log (eq.2)? Can the authors clarify it? Also, the table with fitting parameters for various tempering will be beneficial
  4. There are also missing information for example Table 4 where RA (Reduction of Area) is missing in some cases without explanation.
  5. Figures and their captions should be self-explaining and all necessary information should be provided (for example magnification Fig. 1-3, selection of data in Fig. 6, identification of material Fig. 10 etc.).
  6. If Table 3 corresponds to Fig. 5 is not clear the representation of tl and D parameter – if tl is Time Lag parameter, i.e. onset of the curve determined for curve O one can estimate form graph value about 170000s but in the table is 220000s. Can the author clarify this discrepancy which could be caused by a different definition of tl.
  7. The fractographic analysis should be better discussed as „Some photos were reported in Figure 7, Figure 8 and Figure 9 as an example.“
  8. The last but not least issue are some English terms or sentences used, for example: “elevated mechanical properties”, “mechanical performances”, “Permeation tests and slow strain rate tests were used for comparison, …”, “Permeation tests and slow strain rate 60 tests were scheduled in test plan”, etc.  

Author Response

Response to Reviewer 1:

  • Materials and methods paragraph already contains chemical composition and typical microstructure. A sub-paragraph is dedicated to each test procedure. Major details were added to test procedures. Considering SSRT in particular, the standard references were also mentioned for further details. Discussion paragraph was reviewed to better address literature contest and a conclusion paragraph was written.
  • Dedicated nomenclature description page was included at the end of the document.
  • As reported in the reviewed paper, the fourth polynomial grade was chosen in order to eventually describe an inflection point in strain hardening vs. plastic strain curve. It was already visible for some DP steels in [1] and the third polynomial grade was not able to represent it once derived to get strain hardening function. Table with fitting parameters was added.
  • Foot note was added.
  • Figures 1-3 already presented the scale marker, however it was presented more clearly. Rest of figure captions were adjusted.
  • Diffusion coefficient were effectively calculated according to ISO 17081:2014 standard time lag method. The tangent line needed to determine the time lag parameter should be drawn on the integral function curve and not on the flux plot of figure 6.
  • More detailed discussion was added to fractographic analysis.
  • English spelling and expressions corrections were made.

[1]          Dimatteo, A.; Colla, V.; Lovicu, G.; Valentini, R. Strain Hardening Behaviour Prediction Model For Automotive High Strength Multiphase Steels, Steel Res. Int. 2014, 86, 1574-1582. [https://doi.org/10.1002/srin.201400544].

Reviewer 2 Report

This manuscript investigates anti-HE and SCC performance of supermartensitic stainless steel 16Cr5NiMo by performing the permeation testing and slow strain rate testing. Unfortunately, the presentation does not respect the elementary rules of scientific writing. There are many grammatical errors and missing references in this manuscript. Moreover, some graphs (e.g. Figs. 5, 6 10 and 11) are too small to see. I cannot understand some paragraphs due to these errors and blurry figures and evaluate the validity of this study.

I strongly recommend that the manuscript must be confirmed by all authors before submission. I believe that this manuscript should be reorganized and reformatted according to the Guide for Authors, and then resubmitted to this journal. At this stage, I recommend that this manuscript should be rejected for publications in the scholarly journal.

Author Response

Response to Reviewer 2:

  • English spelling and expressions corrections were made.
  • Some missed literature references were added for completion.
  • Axis titles and scales were expanded for better reading.
  • Paper structure was reviewed according to MDPI instructions for authors

Reviewer 3 Report

Paper deals with important subject of hydrogen embrittlement on the supermartensitic stainless steel. It is interesting, but some details need to be added. Comments I recommend to improve are: 

Abstract needs to be rewritten in order to clearly explain aim of paper. In experimental methods please explain how retained austenite was determined. Statement that retained austenite acts as a trap is not correct entirely. Although in it solubility oh H is significantly higher, interface between martensite and austenite is barrier with high energy. Moreover trapping of H in austenite is not recommended as it can transform to martensite causing flood of microstructure with H. Similar is mentioned in permeation test section. In metallography there is statement about highlights in purple blue carbides. What carbides? In hydrogen charging please state charging current and also charging times. If possible please add amount of H adsorbed into the samples. In discussion more comparison to hydrogen embrittlement is expected. Again here wrong statement is said about hydrogen diffusion pathway. In ferritic (BCC) type steels grain boundaries are traps for hydrogen atoms and doesn't act as highways. Numerous studies on this subject were published. Conclusions weren't presented. Was wrong file uploaded?

There is minor English language editing needed.

Author Response

Response to Reviewer 3:

  • Abstract was reviewed to better explain the aim of the work.
  • Retained austenite fractions were reported in Table 2 according to literature results of [1]. A sentence to better explain it was added.
  • About retained austenite acting as a trapping site, it is clear that thanks to a higher solubility it can trap significantly higher diffusible hydrogen amounts compared to martensite, consequently reducing the steel diffusivity. Also if our intent was not to suggest to use retained austenite as a trapping sites, it is fair to underline that microstructural transformation from austenite to martensite due to plastic flow is quite typical of mechanical laboratory testing. But components in operative life, usually work within the elastic range and no plasticity nor transformation is involved. Obviously, it can be different for parts which work reaching the plastic field just because of austenite to martensite transformation.
  • Carbides were described now in discussion paragraph.
  • About charging times and current densities:
    • Charging time is reported in the TEST condition cell of Table 4 for cathodic protection technique. Differently for test in immersion in NACE A the charging time is necessarily the only SSR testing time.
    • No charging current was applied to test sample. The natural galvanic coupling between steel and magnesium was used (cathodic protection condition). The voltage between anode and cathode in the electrochemical cell was measured and is reported in the text. No data available about circulating current.
    • No measures of diffusible hydrogen content were performed on SSR test samples.
  • Discussion and conclusions paragraphs were rewritten.
  • Grain boundary was indicated as pathways for hydrogen in terms of crack growth path to give rise to the typical intergranular brittle fracture and not simply for hydrogen diffusion.

[1]          De Sanctis, M.; Lovicu, G.; Valentini, R.; Dimatteo, A.; Ishak, R.; Migliaccio, U.; Montanari, R.; Pietrangeli, E. Microstructural Features Affecting Tempering Behavior of 16Cr-5Ni Supermartensitic Steel Metall. Mater. Trans. 2015, 46A, 1878-1887. [https://doi.org/10.1007/s11661-015-2811-x].

Reviewer 4 Report

In their manuscript the authors investigate both stress corrosion cracking and hydrogen embrittlement of supermartensitic steels. Though this topic is of high technical relevance, in its present form the paper is written too sloppy to be published. The authors should be more precise in data presentation and data evaluation, from my point of view. Once this is done, I recommend re-concideration of the manuscript. Details are given below.

1.) Introductory part: The authors investigate martensitic steels. Hence, these steels show a large tendency of hydrogen environmental brittleness owing to the bcc structure of the matrix. The bcc structure favors rapid hydrogen diffusion from the sample surface to crack tips and dislocations irrespective of additional strain induced martensite formation, once first cracks have formed in fatigue conditions. The authors should explain this circumstance more clearly in the manuscript.

2.) Page 3, line 93ff: The austenite fraction with its larger H solubility will only have an on-top impact on the embrittlement of the steels, when it transforms into strain-induced martensite.

3.) Figs. 1, 2, 3, 9: What can be seen in the images? The authors should explain in the figure captions what is shown in the figures. Where are grain boundaries, where are Laves phases, where are transgranular cracks, etc.?

4.) Page 4, line 115: The diffusivity in fcc is lower by several orders of magnitude compared to bcc. The authors should be more precise by giving numbers, here.

5.) Page 4, line 119: Is the sample surface stable against oxidation in H2SO4+As2O3 environment? Does a possible surface oxidation affect the permeation tests?

6.) Page 5, line 138: Which reference electrode has been used?

7.) Page 6, line 174: What do increased transgranular fracture areas mean with respect to the mechanism of embrittlement?

8.) Page 8, line 185: Why have the data been fitted with a polynomial of fourth grade?

9.) Figure 11: The test results seem very similar for as-received and for hydrogen charged material. This hints on the large effect of the bcc matrix on the embrittlement behaviour, while the retained austenites’ impact  is very small.

10.) Page 9, line 210ff: Can the authors quantify the austenites’ impact on the measured effective diffusion coefficients?

Author Response

Response to Reviewer 4:

  • Major details about embrittlement phenomenon and failure mechanisms added in the manuscript.
  • Yes, we agree.
  • Light microscope images were added of red arrows to highlight carbides precipitated at grain boundaries. Presence of Laves phases was not assessed, the relative sentence in the text was deleted.
  • Diffusivity values in BCC and FCC lattices were added with reference to [1]
  • Comment related to surface oxidation was added in the text. A nitrogen purging flux was applied inside the electrochemical cell to avoid any sample surface oxidation.
  • An Ag/AgCl reference electrode was used for measurements.
  • Discussion about failure mechanism was added.
  • As reported in the reviewed paper, the fourth polynomial grade was chosen in order to eventually describe an inflection point in strain hardening vs. plastic strain curve. It was already visible for some DP steels in [1] and the third polynomial grade was not able to represent it once derived to get strain hardening function. Table with fitting parameters was added.
  • No large differences were observed between as received and charged materials, thus the major effect is the embrittlement behavior of BCC matrix. The retained austenite effect was limited, this is also correlated to low values of retained austenite percentages in the microstructure, in fact also between the three steels there were small differences.
  • More details were added in the manuscript. It should be considered that hydrogen diffusivity is certainly directly proportional to retained austenite content and its distribution in the metal lattice [2]. But also martensite tempering temperature and eventual carbides precipitation have no negligible effect. As visible in Table 3 a proportional correlation was found between the amount of retained austenite and effective hydrogen diffusion coefficient.

[1] Woodtli, J.; Kieselbach, R.; Damage due to hydrogen embrittlement and stress corrosion cracking. Engineering Failure Analysis 2000, 7, 427-450. [https://doi.org/10.1016/S1350-6307(99)00033-3]

[2] Olden, V.; Thaulow, C.; Johnsen, R.; Modelling of hydrogen diffusion and hydrogen induced cracking in supermartensitic and duplex stainless steels. Materials and Design 2008, 29, 1934-1948. [https://doi.org/10.1016/j.matdes.2008.04.026]

Round 2

Reviewer 1 Report

The manuscript entitled „Influence of thermal treatment on SCC and HE susceptibility of supermartensitic stainless steel 16Cr5NiMo“ was revised by authors and improved significantly. However, still the present form of the manuscript is not suitable for publication without major changes because not all comments and recommendations were fulfilled.

Main comments and recommendations:

  1. The paper has now the right structure but the content of chapters is not preciously provided. The chapter Materials and Methods does not contain a full description of all experimental methods and parameters used. There still exists rather a literature review or discussion of results subchapters (see rows 77-93 or rows 127-137 for example). The formulation like “high pureness grade” is not scientifically providing good enough information.
  2. Missing information should be updated (see Fig. 4 where no all dimensions are provided, Fig. 6 still suffer by small labels and dots should be used in numbers)
  3. Scatters are not provided for given values at all (Table 4).
  4. Still, it is not clear why authors use polynomial fit with 4th order when in work [27] is used just 3rd order. The higher-order polynomial fits suffer by instabilities (see the beginning of fit in Fig 11b or Fig 12)
  5. Figure Fig. 11 still don’t link with a specific material.
  6. The fractographic analysis contains still the sentence: „Some photos were reported in Figure 8, Figure 9 and Figure 10 as an example.“, it was slightly improved but not enough from my point of view.
  7. The discussion should be based on evidence and no microstructural and phase analysis were in the result part provided. Also, changes in martensite type or amount during deformation were not measured.
  8. The frequent usage of “filling” words (in fact, on the other hand, moreover, indeed ...) is not appreciated.
  9. Wrong cross-references (for example row 202, 249) should be corrected as well as may of typos.

Author Response

Thanks for your suggestions, further revisions have been made according to  reviewer. Please see the attachment.

Reviewer 2 Report

I previously pointed out that the presentation of this manuscript did not respect the elementary rules of scientific writing. Some of them are modified, but the modification is insufficient. For example, in Fig. 6, all letters are too small to read when this manuscript is printed out. Also, in Fig. 11(a) and (b), I do not understand what lines and points represent because there are no values on the vertical and horizontal axes. There are many other errors. Due to these errors related to presentation, I cannot review this manuscript the content of this study.

In my opinion, the modification of this manuscript is unfinished and it cannot pass revision, which is negative. The final decision I leave to the editor.

Author Response

Thanks for your suggestions, we applied a lot of other modifications to meet rules of scientific writing.

Figures 6 and 11 were updated according to indications.

Reviewer 3 Report

All reviewer suggestions were followed and language was significantly improved. 

Author Response

Thanks for your suggestions, we applied few more modifications according to other reviewer requests.

Reviewer 4 Report

The paper was widely rewritten in its main parts and now complies with scientific standards. Important details on HE mechanisms and the expected role of retained austenite in the steels behavior were added, as well as details of permeation tests and data evaluation. From my point of view, only minor questions remain prior to publication.

1.) Permeation tests were performed at RT. Hence, on page 5, line 140 typical RT diffusion coefficients of FCC and BCC steels should be given instead of values at 200°C.

2.) Considering RT standard diffusion coefficients – is it possible to quantitatively compare measured diffusion coefficients in Tab. 3 with the standard values by linking them to the volume fraction and possible distributions of retained austenite traps?

3.) What do we learn from the SEM images in Figs. 1-3? These images are not considered in the text.

4.) As a side remark, to my knowledge there is now experimental evidence for the HEDE mechanism on the atomistic scale, so far.

Author Response

  • In order to get a better comparison, typical diffusivities of ferritic and austenitic lattices at room temperature were found and reported in the text.

  • Microstructures of steels under investigation were different in terms of retained austenite amounts and a difference in hydrogen diffusivity was found. Consequently, confirming other literature studies, hydrogen diffusion coefficient has a certain proportionality with the austenitic phase content. In particular a good exponential correlation was found. Moreover, previous studies about the same materials, by means of TEM, presented the austenite fraction as elongated interlath films in a martensite matrix.

  • Reference to light microscope and SEM images was missing. It was added in correspondence of LINE 113.

  • HEDE was presented within the short description of state of the art. Many authors discuss about it thus we mentioned it. Some references were also linked as support.

Round 3

Author Response

In the present version we cut the ends of the curve, since we recognize they are not well approximated but at the same time they are not useful in the present study. All regressions were recalculated after the cut.

The use of a fourth-grade polynomial and in general a polynomial function, is for sure not the most suitable in term of physical significance. That was a mathematical artifice to reach our main investigation aim on the plastic behavior of materials.

Moreover, we have added some experimental data in fig. 12b in order to be clarify on the goodness of the approximation, and it looks to work.

Reviewer 2 Report

I previously pointed out that the presentation of this manuscript did not respect the elementary rules of scientific writing. They are modified, and I think that this manuscript should be accepted in the present form.

Author Response

We didn't received new comments, so thank you for your collaboration and your suggestions.